# Is active sitting on a dynamic office chair controlled by the trunk muscles?

Roman Peter Kuster[1,2,3,4]*, Christoph Michael Bauer[3], Daniel Baumgartner[2]

**1** Division of Physiotherapy, Department of Neurobiology, Care Sciences and Society, Karolinska Institutet, Stockholm, Sweden, **2** Institute of Mechanical Systems, School of Engineering, Zurich University of Applied Sciences, Winterthur, Switzerland, **3** Institute of Physiotherapy, School of Health Professions, Zurich University of Applied Sciences, Winterthur, Switzerland, **4** Institute of Human Movement Sciences and Sport, Department of Health Sciences and Technology, Federal Institute of Technology, Zurich, Switzerland

* roman.kuster@alumni.ethz.ch

**Data Availability Statement:** All relevant data are within the manuscript.

**Funding:** One of the authors (RK) was partly financed through a personal grant of the Swiss National Science Foundation (http://www.snf.ch/en,

## Abstract

Today's office chairs are not known to promote active sitting or to activate the lumbar trunk muscles, both of which functions are ergonomically recommended. This study investigated a newly developed dynamic office chair with a moveable seat, specifically designed to promote trunk muscle controlled active sitting. The study aimed to determine the means by which the seat movement was controlled during active sitting. This was accomplished by quantifying trunk and thigh muscular activity and body kinematics. Additionally, the effect of increased spinal motion on muscular activity and body kinematics was analysed. Ten subjects were equipped with reflective body markers and surface electromyography on three lumbar back muscles (multifidus, iliocostalis, longissimus) and two thigh muscles (vastus lateralis and medialis). Subjects performed a reading task during static and active sitting in spontaneous and maximum ranges of motion in a simulated office laboratory setting. The temporal muscle activation pattern, average muscle activity and body segment kinematics were analysed and compared using Friedman and post-hoc Wilcoxon tests (p≤0.05). Active sitting on the new chair significantly affected the lumbar trunk muscles, with characteristic cyclic unloading/loading in response to the seat movement. Neither thigh muscle activity nor lateral body weight shift were substantially affected by active sitting. When participants increased their range of motion, the lumbar back muscles were activated for longer and relaxation times were shorter. The characteristic activity pattern of the lumbar trunk muscles was shown to be the most likely dominant factor in controlling seat movement during active sitting. Consequently, the new chair may have a potential positive impact on back health during prolonged sitting. Further studies are necessary to analyse the frequency and intensity of active sitting during daily office work.

## Introduction

Up to 72% of the population in the western world works predominantly in a seated position [1], which exposes a large proportion to the risk of developing sitting-related musculoskeletal complaints. Partially conflicting findings have been reported for the spine and the lower back.

project P1SKP3-187637). The funders had no role in study design, data collection and analysis, decision to publish, or preparation of the manuscript. The other authors received no specific funding for this work.

**Competing interests:** One of the authors (DB) is the inventor of the new seat motion and owner of rotavis AG who holds the right for commercial use of the intellectual property right. The intellectual property right belongs to ETH Zurich which has approved the present study. This does not alter our adherence to PLOS ONE policies on sharing data and materials.

Although a general causal relationship between sitting and low back pain (LBP) has not been confirmed [2–4], prolonged sitting in unfavourable postures [5, 6], as well as static sitting with continuous isometric muscular activity [7, 8], have been found to increase the risk of developing LBP. Sitting-related LBP has been proposed to be caused by prolonged low static trunk muscle activity [7–9], which could lead to deconditioning of the lumbar spine [10]. Patients with LBP typically show an atrophy of the lumbar multifidus [11, 12] and the lumbar trunk muscles are found to be inactive for 30% of the sitting time [10]. As a consequence, patients with LBP and associated spinal problems, e.g. spinal stenosis, disc prolapse and degenerative disc diseases, are known to have a reduced range of motion (RoM) of the lumbar spine [13]. Active motions have been shown to be better for the intervertebral discs and spinal muscles compared to single static postures [6, 14, 15]. Ongoing postural changes result in altered back muscle activity, spinal load and trunk-thigh angle; factors that are thought to be favourable in the prevention of sitting-related LBP, degenerative disc diseases and impaired muscle function [7, 8, 16, 17]. This hypothesis is supported by the fact that 30 to 85% of patients with LBP report that their pain is aggravated by static sitting positions [18, 19] and 34 to 39% report that walking relieves their LBP [18].

Based on these findings, chair manufacturers promoted active sitting. However, conventional dynamic office chairs (Fig 1) have not been found to change a person's sitting behaviour and muscle activity pattern, or to have a positive influence on the development and management of LBP [7, 20–24]. Active sitting typically provokes considerable upper body motion, which is unlikely to be compatible with the demands of working in an office [7, 16]. Subsequently, chair manufacturers began to design instable chairs (Fig 1), with only a small support base to provoke continuous small body motion for maintaining balance [25]. However, these chairs have also not encouraged active sitting behaviour or positively influenced the development of LBP [23, 26–30]. In fact, research findings have shown the opposite effect, contending that subjects seek to maintain balance by keeping their center of mass within the small base of support [27]: a phenomena that has already been observed in standing [31, 32].

For this reason, our group has developed together with rotavis® (Winterthur, CH) a new dynamic office chair. The dynamic principle of instable chairs (which use a motion axis below the seat level to provoke instability) has been inverted and integrated into a conventional office chair (Fig 1). In a previous study, we investigated the boundary conditions for the new seat

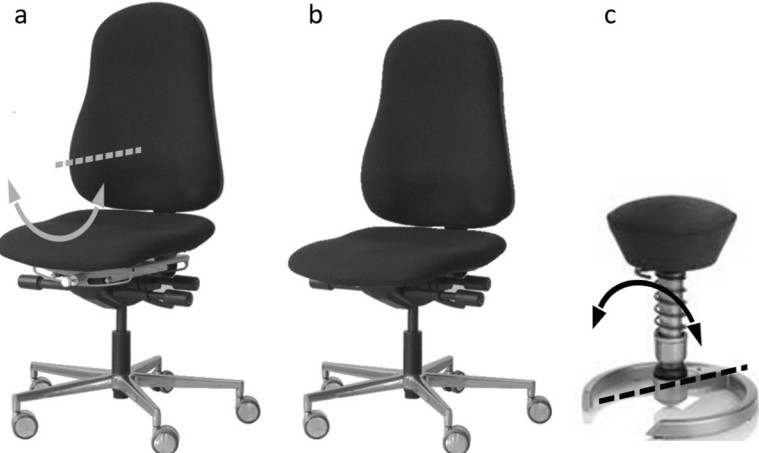

**Fig 1. Office chairs.** The construction of the new chair (a) was based on a conventional dynamic office chair (b) with the inverted dynamic principle of an instable chair (c). In contrast to instable chairs, the seat has a stable central position that rests at the lowest point of the arc corresponding to the centre of the seat.

movements in the frontal plane and found a physiological placement of the motion axis at the level of the 11th thoracic vertebra of the chair user [33]. An additional study, which made a comparison between an instable and a conventional office chair, demonstrated that the upper body remained within a range considered to be stable during active sitting [34], even though subjects performed a substantial lateral spinal flexion. Accordingly, we concluded that active sitting on the new chair is compatible with the demands of office work and that even the backrest can be used during active sitting.

However, the means by which office workers control active sitting on the new chair is still unknown. The primary aim of this study was to quantify the muscular trunk activity, thigh activity and body kinematics of active sitting. If muscular activity contributes to the control of the seat movement, we would expect to find a cyclic unloading/loading of the responsible trunk or thigh muscles. If body weight shift contributes to the control of the seat movement, we would expect to find substantial lateral motion preceding the seat motion. As a secondary aim, the study analysed the effects of increased spinal motion on muscular activity and body kinematics.

## Materials and methods

### Participants

This study was part of a larger study on active sitting and the recordings were made during the second session, described in Kuster et al. 2016 [33]. Ten healthy volunteer office workers (four females, six males) were selected prior to their visit to our motion laboratory. Inclusion criteria were: 1) age between 20 to 50 years; 2) height between 1.53m and 1.92m; 3) BMI <30; 4) working hours of >4 hours per day in a seated position over ≥2 years. Exclusion criteria were: 1) chronic complaints of the back and neck region in the past year; 2) previous spine surgery.

The participants' age was 32.2±7.6 years (mean±SD), body height 1.77±0.09m and body mass 72.1±9.8 kg. Participants worked on average 8.3±0.6 hours a day, of which 6.5±2.1 hours were spent in a seated position, primarily in front of a computer screen. None of the participants had prior experience with the investigated office chair. Ethical approval was granted by the institutional ethics board of the ETH Zurich (EK 2012-N-52) and all participants gave their written informed consent prior to study inclusion. Since no preliminary data were available, a convenient sample of n = 10 was included based on information from similar studies [20, 35].

### Chair

The start-up company rotavis® and our research group equipped a conventional dynamic office chair with an additional degree of freedom in the frontal plane [33]. The additional degree of freedom allowed the seat to slide from side to side on a radius with the centre located above the seat level, approximately at the level of the 11[th] thoracic vertebra of the chair user (Fig 1). Accordingly, the seat was able to rotate in a convex manner around the anterior-posterior axis, resulting in a combined inclination and translation in cranio-lateral direction. Contrary to instable chairs, the seat maintained a stable central position because it rested at the lowest point of the arc corresponding to the centre of the seat. Consequently, no additional muscle activity was needed to keep the seat position centred. The same chair was used for the analysis of both static sitting and active sitting.

### Conditions and experimental procedure

Following completion of the questionnaire on demographics, participants sat on the new dynamic office chair and the seat (90° knee angle) and table height (2 cm above elbow level)

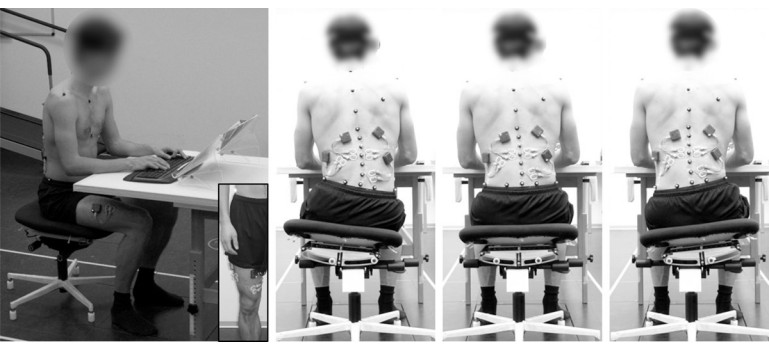

**Fig 2. Measurement setting.** Lateral and posterior views of a subject during active sitting, including electrode and marker placement. The three pictures on the right show examples of the central and the maximum left and right positions. Note that subjects placed their palms on the wrist rest of the keyboard.

were adjusted. They were then equipped with reflective body markers and surface electromyography (sEMG) electrodes. To accustom themselves to the seat movement, participants sat for 1 minute on the chair in an active manner, without receiving movement instructions.

All sitting conditions were recorded while the participants performed a typical reading task (Fig 2). They were instructed to maintain their usual reading performance during the measurement process. With the seat of the office chair fixed in the middle position, sEMG and kinematic signals were recorded for 10 seconds for the static sitting condition (STAT-SIT). For active sitting, three times six motion cycles (seat movement to the left and right side) were recorded as participants used the dynamics of the chair, firstly in a spontaneous RoM (ACTSIT) and subsequently in a maximum RoM (ACTSIT$_{max}$). The motion cycle frequency for ACTSIT was standardized to 0.5Hz using a metronome. This standardization was necessary because users with no prior experience tend to use the chair dynamics at a much lower frequency than experienced users. Only the frequency was prescribed, not the RoM. For sEMG signal reference purposes, following the measurement of the sitting conditions, each participant walked on a treadmill for one minute at 1 m/s. The treadmill was tilted upwards to its maximum (13.5°) gradient to increase motion of the lateral pelvis and spine [33, 36].

## Instrumentation

Muscle activity was recorded using a wireless sEMG system (myon 320; myon®, Schwarzenberg, CH) with a sampling frequency of 1000Hz and a pre-amplification factor of 1000. Pairs of disposable sEMG electrodes (Ambu BlueSensor N) were fixed to the prepared skin of the participants in accordance with the SENIAM guidelines [37]. The electrodes were placed on the longissimus bilaterally and on the right multifidus, left iliocostalis, right vastus medialis and vastus lateralis unilaterally.

Body kinematics were recorded using a 12-camera infrared light-emitting motion capture system (Vicon MX system; Oxford Metrics Group®, Oxford, GB) with a sampling frequency of 200 Hz. Reflective skin markers (12.5mm diameter) were placed on the participants according to the Plug-in-Gait upper body model (Table 1) and reconstructed using the software Vicon Nexus 1.7.1 (Oxford Metrics Group®, Oxford, GB) [38, 39].

Marker placement and segment definition according to the Plug-in-Gait (PIG) upper body model [38, 39], with additional markers on the spine and pelvis. The percentages refer to back length [40]. All PIG markers were used to calculate the centre of mass. Marker placement is also shown in Fig 2.

**Table 1. Marker placement.**

| Segment | Marker | Placement | |
|---|---|---|---|
| Head | l/r-FHD [PIG] | over the l/r temple | |
| | l/r-BHD [PIG] | l/r back of the head | |
| Shoulder | l/r-SHO [PIG] | l/r acromio-clavicular joint | |
| Thorax | C7 [PIG] | 7th cervical spinous process (0%) | |
| | T10 [PIG] | 10th thoracic spinous process (54.5%) | |
| | CLAV [PIG] | between articuli sterno-clavicularis | |
| | STRN [PIG] | xiphoid process of the sternum | |
| Spine | T4 | 4th thoracic spinous process (21.2%) | |
| | $L_{low}$ | centred between L4 and SACR (93.2%) | |
| | SACR [PIG] | centred between l-PSI and r-PSI (100%) | |
| Pelvis | l/r-PSI [PIG] | l/r posterior superior iliac spine | |
| | l/r-SIDE | centred between l/r-PSI and l/r-ASI | |
| Seat | SEAT | middle rear of the seat | |
| **Vectors** | **Origin** | | **Direction** |
| Thorax | midpoint of C7/CLAV | | midpoint of T10/STRN |
| u/l-Spine | C7/ $L_{low}$ | | T4/ SACR |
| Pelvis | midpoint of l-PSI/l-SID | | midpoint of r-PSI/r-SID |

Abbreviations: left and right (l/r), Front of the Head (FHD), Plug-in-Gait (PIG), Back of the Head (BHD), Shoulder (SHO), Cervical (C), Thoracic (T), Clavicle (CLAV), Sternum (STRN), Lumbar (L), Sacrum (SACR), Posterior superior iliac spine (PSI), Anterior superior iliac spine (ASI), upper and lower (u/l).

## Data processing

Data processing was carried out with MATLAB version 8.3 (The MathWorks Inc.®, Natick, USA). The raw sEMG signals were bandpass filtered (Butterworth 2nd order, 50 to 400Hz), rectified, smoothed over a 0.1s window and expressed in relation to the peak activity in walking. To calculate the latter, the maximum of 12 consecutive double steps was determined, averaged and set to 100%.

For STATSIT, the median muscular activity was determined over the middle 2 seconds of the recording. The median was taken because data were not normally distributed (verified with Lilliefors test). For ACTSIT and ACTSIT$_{max}$, sEMG and kinematic data were divided into the individual motion cycles in order to analyse the middle four of each recording, resulting in a total of 12 cycles per subject and condition. The average muscular activity was calculated for each subject and condition. To compare the temporal activity profiles of ACTSIT and ACT-SIT$_{max}$ with STATSIT, the time in which the sEMG amplitude exceeded, was equal to, or fell below the 95% range of STATSIT was analysed and expressed as a percentage of the time.

To analyse upper body motion, the midpoint of the thorax segment was calculated, the distance between its maximum left and right positions determined and then divided by two to express the RoM of a unidirectional seat movement to one side. The RoM of the centre of mass was calculated according to the Plug-in-Gait model and evaluated in the same way [39]. Since the extremities did not move (feet placed on the floor, hands on the keyboard), they were not considered in the centre of mass determination. The angular RoM of thorax and pelvis were analysed in the same way but using directional vectors (Table 1) instead of the midpoint. The lateral flexion of the spine was determined by calculating the angular difference between the uppermost (C7 to T4) and lowermost ($L_{low}$ to SACR) spinal segments and

analysed similarly at the left and right maximum positions. The average movement speed was calculated by expressing the RoM in relation to the movement cycle duration.

Finally, sEMG, as well as centre of mass, thorax and seat movement, were time-normalized to 200 data points (100 for the movement to each side) using a linear interpolation, averaged over all repetitions and subjects and plotted against time.

### Outcome measure

To quantify active sitting, the average muscle activity and its temporal activation pattern were compared to static sitting. The body segment kinematics of the spine, the thorax, the pelvis and the centre of mass were analysed in terms of RoM and movement speed. Additionally, muscle activity and thorax, centre of mass, and seat movements were plotted in relation to the movement cycle. The movement cycle started and ended when the seat was in the central position and included the seat movements to the left and right side. Thus the movements to both sides were analysed.

### Statistics

All statistics were calculated using SPSS 23 (IBM Corp., Armonk, USA). Data normality was tested with Lilliefors test and non-parametric statistics was used for positive test outcomes.

To compare sEMG amplitudes between the sitting conditions, a Friedman ANOVA was used. Significant effects were tested by a post hoc Wilcoxon matched-pair test to compare the individual conditions. The time in ACTSIT and ACTSIT$_{max}$ spent below, equal to and above the muscle activity of STATSIT was expressed with the mean ±SD (normal distribution not rejected). Kinematic data of ACTSIT and ACTSIT$_{max}$ were compared using a Wilcoxon matched-pair test. Since no condition effects were found (tested with a Wilcoxon matched-pair test), movement speed was averaged over ACTSIT and ACTSIT$_{max}$. A p-value ≤0.05 was considered significant and significant effects were additionally described with a 95% confidence interval of the effect.

## Results

### Temporal pattern

The temporal muscular activity and kinematic motion pattern over the entire movement cycle is plotted in Fig 3. The figure shows a cyclic unloading/loading of all the investigated trunk muscles for both active sitting conditions. During the movement to the right side (highlighted in Fig 3), lumbar trunk muscles on the right side of the body contracted, while those on the left side relaxed. The movement to the left side was characterised by the reverse activity pattern. For the thigh muscles, findings showed no cyclic unloading/loading in ACTSIT and a very attenuated unloading/loading pattern in ACTSIT$_{max}$. The centre of mass showed an attenuated and oppositely-directed movement pattern compared to the seat, with no relevant changes when changing the movement direction (at 50% and 150% in Fig 3). The thorax moved along with the centre of mass, but with a slightly larger RoM. It also showed a very consistent course.

### Muscular activity in static and active sitting

An overall condition effect was found for the average muscle activity of multifidus and iliocostalis (Table 2). The post-hoc analysis revealed a significantly increased iliocostalis activity compared to STATSIT for ACTSIT, while for ACTSIT$_{max}$ iliocostalis and multifidus activity were significantly increased compared to STATSIT. The average thigh muscle activity was not affected by active sitting.

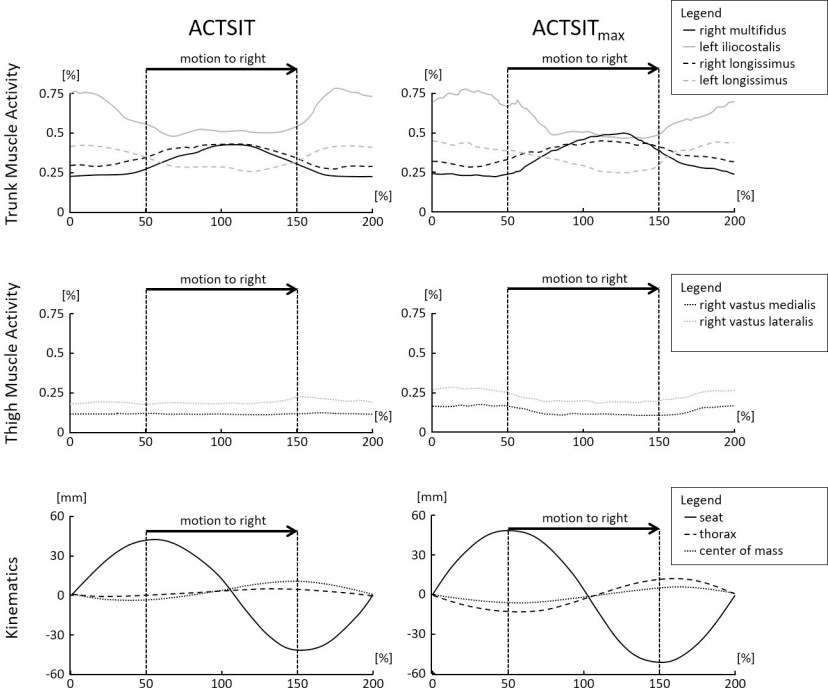

**Fig 3. Temporal activity and motion pattern.** Time normalized muscular activity and kinematic motion pattern during spontaneous (ACTSIT) and maximum active sitting (ACTSIT$_{max}$) over all subjects. The seat is at 0%, 100% and 200% in the central position and at approximately 50% (left) and 150% (right) in the extremal positions. The muscle activity is given as a percentage of the peak activity in walking, kinematics in millimetres (mm).

The temporal activity pattern for ACTSIT and ACTSIT$_{max}$ in relation to STATSIT is shown in Table 3. The proportion of the time below, equal to and above STATSIT muscle activity was approximately equal for longissimus on both sides of the body. The muscle activity of multifidus and iliocostalis were higher during active sitting compared to static sitting for $\geq$49% of the time. The two thigh muscles were equally activated during active sitting and static sitting for most of the time.

## Kinematics in active sitting

The kinematic comparison between ACTSIT and ACTSIT$_{max}$ is shown in Table 4. Increasing the seat RoM significantly affected all the investigated RoMs, except thorax inclination. Since

**Table 2. Average muscular activity.**

|  | STATSIT | ACTSIT | ACTSIT$_{max}$ | p-value | Difference to STATSIT | |
|---|---|---|---|---|---|---|
|  |  |  |  |  | ACTSIT | ACTSIT$_{max}$ |
| **Trunk Muscles** |  |  |  |  |  |  |
| left longissimus | 25.9 [19.4] | 31.6 [10.8] | 31.2 [17.0] | 0.497 |  |  |
| right longissimus | 34.2 [17.9] | 31.9 [18.1] | 37.4 [17.7] | 0.882 |  |  |
| multifidus** | 25.9 [12.3] | 28.4 [15.5] | 30.5 [12.5]** | 0.003 |  | 3.6 [1.1–7.2] |
| iliocostalis** | 43.2 [27.1] | 49.5 [38.1]** | 48.3 [38.9]** | 0.005 | 5.2 [1.5–23.9] | 6.2 [-0.2–18.0] |
| **Thigh Muscles** |  |  |  |  |  |  |
| vastus medialis | 10.7 [7.0] | 11.0 [7.3] | 10.7 [6.3] | 0.165 |  |  |
| vastus lateralis | 12.7 [7.0] | 13.0 [23.8] | 12.8 [25.3] | 0.247 |  |  |

Median muscular activity [inter-quartile range] during static sitting (STATSIT) and active sitting in spontaneous (ACTSIT) and maximum range of motion (ACTSIT$_{max}$), expressed as a percentage of peak activity in walking. Significant effects marked with asterisks (**:p$\leq$0.01). For significant effects, the last two columns give the median difference with 95% Confidence Interval.

**Table 3. Temporal muscular activity pattern.**

| | Trunk Muscles | | | | Thigh Muscles | |
|---|---|---|---|---|---|---|
| | **left longissimus** | **right longissimus** | **multifidus** | **iliocostalis** | **vastus medialis** | **vastus lateralis** |
| **ACTSIT** | | | | | | |
| Time below | 30.6 ±19.1 | 30.6 ±19.4 | 21.8 ±24.9 | 8.6 ±8.4 | 13.9 ±8.1 | 17.4 ±4.0 |
| Time equal | 28.7 ±17.7 | 35.5 ±20.4 | 28.7 ±18.2 | 34.5 ±15.1 | 65.8 ±13.0 | 60.9 ±7.0 |
| Time above | 40.7 ±26.8 | 34.0 ±22.7 | 49.4 ±20.0 | 56.9 ±20.4 | 20.2 ±6.9 | 21.7 ±8.0 |
| **ACTSIT$_{max}$** | | | | | | |
| Time below | 29.2 ±17.0 | 25.6 ±15.7 | 16.6 ±15.8 | 9.1 ±13.0 | 14.1 ±8.4 | 12.1 ±6.1 |
| Time equal | 26.6 ±12.5 | 35.2 ±17.1 | 28.7 ±14.4 | 27.5 ±11.8 | 52.9 ±8.0 | 51.2 ±14.0 |
| Time above | 44.2 ±23.9 | 39.2 ±20.5 | 54.8 ±12.2 | 63.3 ±18.1 | 33.1 ±15.3 | 36.7 ±19.0 |

Indicated is the percentage of the time for spontaneous (ACTSIT) and maximum active sitting (ACTSIT$_{max}$) below, equal to and above the 95% range of static sitting (mean ±SD).

there was no effect of the active sitting condition on the movement speed, speed data in Table 4 are averaged across ACTSIT and ACTSIT$_{max}$.

## Discussion

The aim of this study was to investigate a dynamic office chair with a special moveable seat and its effect on muscular activity and upper body kinematics during active sitting in a simulated office laboratory environment. We hypothesised, based on theoretical considerations, that three mechanisms could be used to control the movement of the seat: a periodic activation of the trunk muscles; a periodic activation of the thigh muscles; and/or a substantial lateral shift in body weight.

### How do office workers control the seat movement?

In the muscular dimension, an overall statistical effect for multifidus and iliocostalis muscular activity between the static sitting and the two active sitting conditions was found. Iliocostalis

**Table 4. Range of motion and movement speed.**

| | ACTSIT | ACTSIT$_{max}$ | p-Value | Difference |
|---|---|---|---|---|
| **Range of Motion** | | | | |
| Spine (Lateral Flexion) [°] | 9.6 [6.5] | 13.4 [3.7]** | 0.007 | 3.8 [0.8–8.2] |
| Thorax (Inclination) [°] | 0.8 [1.0] | 1.4 [0.9] | 0.139 | |
| Pelvis (Inclination) [°] | 6.4 [1.9] | 8.4 [0.6]** | 0.009 | 1.9 [0.0–4.7] |
| Centre of Mass (Translation) [mm] | 4.8 [4.3] | 8.1 [5.7]** | 0.009 | 2.2 [0.0–5.4] |
| Thorax (Translation) [mm] | 7.8 [4.8] | 11.8 [7.4]** | 0.009 | 4.2 [0.4–8.7] |
| **Movement Speed** | | | | |
| Spine (Lateral Flexion) [°/s] | 37.6 [19.4] | | | |
| Thorax (Inclination) [°/s] | 4.2 [1.5] | | | |
| Pelvis (Inclination) [°/s] | 25.0 [5.0] | | | |
| Centre of Mass (Translation) [mm/s] | 19.9 [18.3] | | | |
| Thorax (Translation) [mm/s] | 29.6 [19.2] | | | |

Median range of motion and movement speed [interquartile range] of the investigated body segments in spontaneous (ACTSIT) and maximum active sitting (ACTSIT$_{max}$). Since there was no difference in movement speed between ACTSIT and ACTSIT$_{max}$, data were pooled. The difference between ACTSIT and ACTSIT$_{max}$ is indicated by the median [95% Confidence Interval] in the case of a significant effect.

muscular activity was increased for both active sitting conditions compared to static sitting, while multifidus muscular activity was only increased for $ACTSIT_{max}$ (Table 2). The temporal analysis in Table 3 also showed that both multifidus and iliocostalis muscular activities were significantly increased or decreased during active sitting compared to static sitting for most of the time. The longissimus muscular activity, the third lumbar trunk muscle under investigation, showed no overall condition effect and a more balanced temporal activity pattern compared to static sitting. Thus, it seems that the movement of the seat is primarily effected by the activity of iliocostalis and multifidus, supported by longissimus (Fig 3). This conclusion corresponds to our expectations based on the anatomical muscle orientation. While longissimus has the most cranially directed fibre orientation, multifidus and iliocostalis have a more laterally directed fibre orientation that is likely to be more efficient in generating the investigated lateral seat movement. Accordingly, iliocostalis showed the highest activity compared to walking. Nevertheless, all recorded lumbar trunk muscles showed the expected temporal variation. During the right movement of the seat (from left to right maximum seat position, highlighted in Fig 3), we observed increased activity of the right multifidus and right longissimus, together with decreased activity of the left iliocostalis and left longissimus. The movement to the left side was characterized by the reverse pattern. The average thigh muscle activity was not affected by active sitting (Table 2) and remained within the same activity range for static sitting most of the time (Table 3). Thus, the thigh muscle is unlikely to have caused the seat movement. However, the results of the temporal analysis in Fig 3 show a slight variation in the thigh muscle activity level for $ACTSIT_{max}$. This indicates that participants supported the trunk muscles with the thigh muscles to some degree to increase the RoM.

If the lateral body shift controls the seat movement, we would have expected to find a substantial displacement of the centre of mass or the thorax, or at least a body sway preceding the lateral seat motion. However, the centre of mass RoM in ACTSIT (4.8 mm) lay below the centre of mass sway limits reported for a typical typing task (5.3 to 6.9 mm, [27]). Also, Fig 3 shows that the centre of mass path did not precede the seat movement. For ACTSITmax, the centre of mass path even seems to pursue the seat movement. This makes the centre of mass movement very unlikely to have caused the seat movement. The same is true for the thorax movement, which also did not change substantially (0.8 mm in ACTSIT) or precede the seat movement (Fig 3).

Due to the stable upper body posture, the observed trunk muscle activity pattern is likely to have been the dominant factor controlling the seat movement. The trunk muscles are not needed to stabilise the upper body, because the upper body remains stable and upright during active sitting.

### How does active sitting affect the trunk muscle activity?

When comparing active sitting with static sitting, only a small increase in the average muscular activity amplitude was observed. This means that the increase in muscular activity during the movement to one side was compensated by a decreased activity of the same muscle during the movement to the other side. Consequently, the average muscular load was not affected by active sitting, except for iliocostalis that appeared to be the most important muscle in controlling the seat movement. When the range of motion was increased, a similar effect was also noticed for multifidus. The temporal analysis of all trunk muscles clearly showed a substantial variation in muscular activity (Fig 3 and Table 3), as ergonomically recommended for the prevention of sitting-related LBP [7, 8, 16, 17]. For this reason, when the movement is performed regularly during everyday office activities, the cyclic variation of lumbar back muscle activity is hypothesized to counteract the atrophy found in patients with LBP [11, 12], as well as spinal

deconditioning [10]. However, the intensity and frequency required to cause an effect for lumbar back muscles, as well as the feasibility of implementation during daily office work, remains a subject for future investigations.

## How does active sitting affect the trunk kinematics?

To quantify the kinematic activity during active sitting, we present the movement speed of the various body segments. Unfortunately, a direct comparison between active sitting and walking was impossible for us because our treadmill had to be placed outside the measurement area of our motion capture system. However, a comparison to data on walking from existing literature appears to be useful because both active sitting and walking are performed with a very similar pelvis RoM [33, 36, 41, 42]. The observed speed of the lateral spine flexion (38°/s) and pelvis inclination (25°/s) were clearly higher than those reported for walking (lateral spine flexion: 27–33°/s, pelvis inclination: 10–21°/s [36, 41, 42]). A similar movement speed can be found in double step stair ascent (lateral spine flexion: 36°/s, pelvis inclination: 26°/s [36]). However, the thorax inclination (4°/s) was slower than in walking (6°/s [43]), meaning that although the spine and pelvis moved faster, the thorax remained more stable. These observations are in line with the aim of our new chair: to promote trunk controlled active sitting while maintaining upper body stability, so that office workers can focus on the work task. Due to the very similar upper body kinematics of active sitting and walking, a future study should analyse whether patients with LBP, whose back pain is relieved by walking, would also benefit from active sitting [18]. If so, the chair might allow patients with LBP to get pain relief without interrupting their office work.

Participants significantly increased their lateral spine flexion (+3.8°) with expansion of the range of motion, through increasing pelvis inclination and, to a smaller extent, thorax inclination, whilst keeping the same speed of movement. Despite this, the thorax movement remained ≤12 mm and ≤1.4°, within the range observed in real office situations using conventional dynamic office chairs (30 mm translation [7]); 2.8° lateral inclination [20]. Comparing the results of this study (centre of mass speed of 20 mm/s) to those in Grooten et al. 2013 (centre of pressure speed of 27 mm/s for standing and 45 mm/s for a conventional office chair) implies that our active sitting condition was less active, even though we observed a lateral spine flexion faster than in walking [27]. We therefore recommend future studies to be cautious when using the centre of mass path or speed as a measure of activity. Even the difference in methods, kinetic in Grooten et al. 2013 [27] versus kinematic (this study), does not explain the difference in conditions with such low centre of mass speeds (<50 mm/s; [44]). The authors of another study [45] observed a difference between centre of mass and centre of pressure of about 35 mm in a standing balance task of 30 seconds that explains differences of up to 1.2 mm/s. We agree that the centre of mass and centre of pressure are useful for quantifying stability [46, 47], but not for quantifying activity however. Active sitting on the new chair was found to be stable (slow centre of mass motion), but highly active (fast lateral spine flexion). Our recommendation, in general, is to evaluate segmental movement speeds in order to quantify the kinematic activity level. Moreover, we recommend analysing the energy expenditure during active sitting on the new chair in order to quantify the activity level of active sitting for public health purposes (sedentary behaviour).

## Critical appraisal

This study has carefully considered some critical issues. The study included only ten office workers, who were unfamiliar with the dynamic chair. To investigate their spontaneous active sitting behaviour, they were given only a short time to familiarise themselves with the chair.

Although the sample size is similar to that of previous studies in this field of research [20, 35], it limits the generalisability of the observed results and may partially explain the quite large variation observed in the presented results. In a subsequent study, it would be of great interest to analyse whether office workers who are already familiar with the new seat motion show a different muscular activity pattern. Furthermore, it would be very interesting to repeat this study with a novel dynamic chair that has a second additional degree of freedom in the sagittal plane [48]. Such a future study should use the data presented in this study to calculate the sample size, in order to draw more generalised conclusions.

This study focused on three superficial back muscles to analyse trunk muscle activity. Since previous studies had reported no differences between body sides [25, 49, 50], only one muscle was analysed bilaterally: the findings for the left and right longissimus in this study confirm the previous results. We considered the lumbar back muscles investigated in this study to be the most important for trunk stabilization in sitting [20, 51] and for the management of LBP [11, 12]. We were also limited to the study of superficial muscles due to our use of sEMG sensors. It would be of great interest to analyse additional muscles using invasive EMG techniques, e.g. the role of the iliopsoas [11]. Unfortunately, sEMG data for iliocostalis and r-longissimus were not available for 1 and 2 participants, respectively. Since no difference between r- longissimus and l- longissimus was found, we assume that this did not influence the results.

To compare sEMG data on the study population level, we referenced data to the average peak value observed in upwards walking. Other studies have referenced sEMG data to a (sub) maximum voluntary contraction [20, 27, 49, 51] or to a standardized posture [10]. However, due to the lack of standardisation of experimental protocols, comparison of sEMG data from different studies is limited, even if the same reference method is used [52]. It is also debatable whether sEMG data gathered from dynamic movements can be referenced to an isometric contraction performed in a different body posture [53]. For these reasons, we referenced our sEMG data to another well-known dynamic motion with similar lateral spine flexion that met our expectation for active sitting (stair ascent [33, 36]), although this limits the comparability of the reported values to other investigations. For practical reasons, we finally decided on a treadmill at its maximum inclination (13.5˚). It must be noted that these reference values had no effect on the statistical analysis conducted in this study. However, the absolute values presented in Table 2 would have been lower if a reference with higher activity was taken (e.g. maximum voluntary contraction).

The step detection for walking was performed manually using video recordings because the treadmill was situated outside the measurement area of our motion capture system. Consequently, we were unable to compare the kinematics of active sitting and walking. This is the reason that we have discussed the movement speed in sitting compared to the literature data on walking. Corresponding to the active sitting conditions, we also used 12 double steps to calculate the average peak value in walking. Shiavi et al. 1998 demonstrated that 6 to 10 cycles are sufficient to produce a representative activity pattern [54]. We concluded from this data that the investigated 12 repetitions per condition were entirely sufficient to study the muscular and kinematic activity for both active sitting and walking. All sEMG data were analysed at their recorded time scales [53, 55] and we only used time normalization for the visual presentation of the results (Fig 3). Lastly, we analysed the lateral body motion only with markers placed on the torso, with no consideration of the extremities. With feet on the floor and hands on the keyboard, the lateral torso motion is a very close approximation to the lateral whole-body motion. It is very unlikely that non-recorded leg or arm movements were used to control active sitting. In conclusion, we are confident that the above-mentioned limitations to this study are fully acceptable when answering the proposed research questions.

## Conclusion

This study shows that active sitting on the investigated office chair is primarily controlled using a very characteristic activity pattern of the lumbar trunk muscles. Active sitting requires a cyclic unloading/loading of iliocostalis, multifidus and, to an attenuated degree, of longissimus. Neither thigh muscle activity nor lateral shift of body weight were substantially affected by active sitting. It is proposed, therefore, that the new chair could have a positive impact on back health during prolonged sitting activities. A future study should further investigate the use of the chair in a real office setting under normal working conditions.

## Acknowledgments

The authors acknowledge Sarah Oetiker for her valuable support in the motion laboratory and Karen Linwood-Williams for the language review.

## Author Contributions

**Conceptualization:** Roman Peter Kuster, Christoph Michael Bauer, Daniel Baumgartner.

**Data curation:** Roman Peter Kuster.

**Formal analysis:** Roman Peter Kuster, Christoph Michael Bauer.

**Investigation:** Roman Peter Kuster.

**Methodology:** Roman Peter Kuster, Christoph Michael Bauer, Daniel Baumgartner.

**Project administration:** Christoph Michael Bauer, Daniel Baumgartner.

**Resources:** Daniel Baumgartner.

**Supervision:** Daniel Baumgartner.

**Writing – original draft:** Roman Peter Kuster.

**Writing – review & editing:** Roman Peter Kuster, Christoph Michael Bauer, Daniel Baumgartner.

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
