## [Decision Letter · Decision Letter 0]

22 Jan 2020

PONE-D-19-28644

Active sitting on a dynamic office chair - Is it trunk muscle controlled?

PLOS ONE

Dear Mr Kuster,

Thank you for submitting your manuscript to PLOS ONE. After careful consideration, we feel that it has merit but does not fully meet PLOS ONE’s publication criteria as it currently stands. Therefore, we invite you to submit a revised version of the manuscript that addresses the points raised during the review process.

We would appreciate receiving your revised manuscript by Mar 07 2020 11:59PM. To enhance the reproducibility of your results, we recommend that if applicable you deposit your laboratory protocols in protocols.io, where a protocol can be assigned its own identifier (DOI) such that it can be cited independently in the future. For instructions see: http://journals.plos.org/plosone/s/submission-guidelines#loc-laboratory-protocols

We look forward to receiving your revised manuscript.

Kind regards,

Juliane Müller, PhD

Academic Editor

PLOS ONE

Journal Requirements:

I have read the journal's policy and the authors of this manuscript have the following competing interests: One of the authors (DB) is the inventor of the new seat motion and owner of rotavis AG who holds the right for commercial use of the intellectual property right. The intellectual property right belongs to ETH Zurich which has approved the present study.

Additional Editor Comments:

The study addresses a very interesting and highly relevant topic. Clearly, the authors are experts in the field and have conducted a sound experiment. Nevertheless, there are some major issues that needs to be revised before consideration for publication in PLOSOne. Please carefully edit the manuscript and copy-edit it prior to resubmission. Please see the specific details within the reviewer comments.

Reviewers' comments:

Reviewer's Responses to Questions

**Comments to the Author**

1. Is the manuscript technically sound, and do the data support the conclusions?

Reviewer #1: Partly

Reviewer #2: Yes

2. Has the statistical analysis been performed appropriately and rigorously? 

Reviewer #1: No

Reviewer #2: Yes

3. Have the authors made all data underlying the findings in their manuscript fully available?

Reviewer #1: Yes

Reviewer #2: Yes

4. Is the manuscript presented in an intelligible fashion and written in standard English?

Reviewer #1: No

Reviewer #2: Yes

5. Review Comments to the Author

Reviewer #1: Manuscript PONE-D-19-28644

Title: Active sitting on a dynamic office chair - Is it trunk muscle controlled?

General Comments

The goal of this study was to investigate a new dynamic office chair with moveable seat developed to promote trunk muscle controlled active sitting. The study aimed to determine the strategy by which subjects control seat motion during active sitting (through trunk or thigh muscle activity, or by a lateral shift in body weight), to quantify muscular activity and body kinematics during active sitting, and to analyse the immediate effect of a spinal range of motion training on muscular activity and body kinematics during active sitting. While the study addresses interesting questions the following major shortcoming of this study require additional information and revisions.

Major comments

1. Although the authors state that they consulted assistance for writing the manuscript, the manuscript contains several grammatical errors and some sentences are difficult to understand. The manuscript should be carefully copy-edited prior to resubmission.

2. The study design is not fully suitable for answering the study objective as the mechanism of controlling sitting on the dynamic chair cannot be experimentally elucidated. Nonetheless, observed difference in study parameters between conditions are highly relevant. The conclusions should be revised accordingly.

3. The authors postulate that there are three different mechanisms for achieving seat motion. Why is it either or? It is also possible that it is a combination of these mechanisms. How can you unlink these mechanisms? Trunk muscle activity is necessary to initiate trunk motion.

4. The authors state in their second objective that the study quantifies active sitting on the new chair regarding muscular activity and body segment kinematics, and analyses the immediate effect of a spinal RoM training on muscular activity and body segment kinematics. This is the main aim of the study. The first objective is a mere interpretation of these results and would be appropriate in the discussion section. To proof these interrelationships would necessitate a musculoskeletal model where each of the stated factors is modulated systematically.

5. The authors may consider restructuring their manuscript. For instance, shouldn’t the outcome parameter be the result of the data processing? Please rearrange the methods section to follow a logical flow.

6. The authors repeatedly mention spinal ROM training yet miss to describe what they mean. Moreover, this study did not involve any training and hence all statements regarding training should be removed.

Minor Comments

1. Abstract: The objective should be rephrased as specified in the comments to the introduction section.

2. Abstract: The conclusion is not supported by the data presented in this study. See also comments in the introduction and discussion sections.

3. Line 43: to develop -> of developing

4. Line 45: colons are followed by lower case. While -> while. Please check throughout the manuscript as this mistake is repeated several times in the text.

5. Line 47: to develop -> of developing

6. Line 49: Better: This may lead to deconditioning of lumbar spine muscles?

7. Line 49: LBP patients. Not defining patients by their disease is more respectful of the person. Please change to patients with LBP throughout the manuscript.

8. Line 65: having only -> with only

9. Line 66: in order to -> to

10. Line 67 and entire manuscript. Please carefully review the literature - for instance, this paper is missing: Effects of a Dynamic Chair on Chair Seat Motion and Trunk Muscle Activity during Office Tasks and Task Transitions.

11. Line 76-77: “…that the upper body remains during active sitting within a range considered to be stable” -> “…that during active sitting the upper body remains within a range considered to be stable”

12. Line 81: it was so far not known -> to date it is unknown

13. Line 82 and entire manuscript: carefully check the tense. The purpose statement should be in past tense.

14. Line 82: to uncover -> better: to determine

15. Line 82-84: Why is it either or? It is also possible that it is a combination of these mechanisms. How can you unlink these mechanisms? Trunk muscle activity is necessary to initiate trunk motion.

16. Line 83: latter – this is not the correct term. The intended meaning is unclear.

17. Line 92: Opposed to instable chairs -> In contrast to other instable chairs

18. Line 93: without muscular effort – without a user? What is your intended meaning?

19. Line 97: Participants of this study…. -> This study was part of a larger study on….

20. Line 98: The – delete

21. Line 99: Please include inclusion and exclusion criteria for the participants of this study.

22. Line 100: Please provide SI units -> m

23. Line 100: body weight -> body mass

24. Line 101: better: … of which 6.5 +/- 2.1 hours were spent in a seated posture

25. Line 102: Better: None of the participants had any experience with the investigated office chair prior to the study.

26. Line 103: by institutional -> by the institutional

27. Line 106: Please provide more information on sample size calculation.

28. Line 110: The additional degree…. – phrase differently: slide side to side on a radius with a centre…

29. Line 114: better: … because it will rest at the lowest point of the arc corresponding to the centre of the seat…

30. Line 115: better: …no additional muscle activity should be needed to…

31. Line 116. Split sentence: The same chair…

32. Line 118: start with: Muscle activity was recorded using…

33. Line 121: Please include citation for the seniam guidelines.

34. Lines 122ff: There is no reason for abbreviating muscle names. These abbreviations make the text more difficult to read and should be removed throughout the manuscript. Moreover, please use lower case for all muscle names consistently throughout the manuscript, e.g. multifidus muscle….

35. Line 124: start with: Body kinematics were recorded using…

36. Line 126: reflective body markers -> reflective skin markers

37. Line 127: Please provide information on marker size and reference to the PIG model.

38. Table 1: Please define all abbreviations in the table footnotes.

39. Line 132: on spine -> on the spine

40. Line 134: additionally -> also

41. Line 140: was -> were

42. Line 141: to left -> to the left (did you only analyse seat motion from left to right?)

43. Line 143: average muscle activity: here, data processing was not explained yet. The average EMG signal will be close to 0. I assume you full-wave rectified the data and then computed the average? Please make sure that this is explained in the right order.

44. Line 146: The authors may consider moving this section to right after the chair description.

45. Line 150: one minute -> 1 minute (numbers with units time always as digits)

46. Line 151: better: Participants did not receive any movement instructions.

47. Line 154: Muscular -> sEMG

48. Line 54: static sitting condition: Was the chair fixed in this position? Please clearly concisely the experimental conditions.

49. Line 168: How do you know that this is the optimal motion frequency?

50. Line 161: When was this done? Before or after the seating conditions? Please specify.

51. Line 165: Lateral and back view on… -> Lateral and posterior view of…

52. Line 167: extremal seat positions -> maximum left and right positions

53. Line 176: why did you use median and not mean?

54. Line 177: data was -> data were (data is the plural of datum)

55. Line 181: statistical significance? This procedure is not clear. One should always choose clinical relevance over statistical significance.

56. Line 183: body segment thorax -> thorax segment

57. Line 184: two maximum positions (to left and right) -> its maximum left and right positions

58. Lien 184: halved -> divided by two

59. Line 185: Was the movement symmetrical to both sides or did some subjects move more to one side than the other? This may influence muscle activity. The subject in Figure 2 has more spine curvature to the left than the right. It seems that this subject is not placed in the centre of the seat. This could largely influence the data. How did you control for this?

60. Line 191: How were the spinal segments defined? Please specify.

61. Line 199: against normality -> for normality (Please also specify the test you used to test for normality)

62. Lines 201ff: The statistical approach only addresses the second objective. How did you address the first objective?

63. Line 210: Figure 3 was unreadable – and hence I cannot comment on the data presented here.

64. Line 229: Please do not use abbreviated muscle names as stated above.

65. Line 230: significantly higher ILIO activity – compared to what?

66. Line 232: Please provide statistical results of the post-hoc tests, ideally as 95% confidence interval of the difference.

67. Table 2: How do you explain the large variability in the data presented in this table?

68. Line 241: MULT and ILIO were…. – please rephrase this and the subsequent sentence as the intended meaning is not clear.

69. Line 248: Did the difference in left and right muscles occur during the same part of the activity? Why is the 50% mark important? Please show muscle activity patterns. Greater muscle activities in periods of greater muscle activity would be functionally more relevant.

70. Line 253: data in Table 4 is -> data in Table 4 are

71. Table 4: Pleas report the 95% confidence interval of the difference.

72. Line 264: What about a combination of these mechanisms? See also comment to the introduction section.

73. Line 266: better: In particular, ILIO (don’t use abbreviation) was increased for both active sitting conditions compared to…

74. Lines 268-273: Please do not repeat results in the discussion section. It is ok to summarize the results at a very high level.

75. Line 277: preferred -> more efficient?

76. Line 282: by a vice versa activity pattern -> by the opposite pattern?

77. Line 283: How do you know that it is not the result of the motion? Additional activity might be necessary to stabilize the trunk when it is off centre.

78. Line 284: This may be because the thigh muscle activity is much lower during sitting than during walking. Again, here one would need to see the muscle activity patterns because a very small impulse might be sufficient for initiating the motion.

79. Line 288: One would need to look at the timing of the activity. Unfortunately, Fig 3 is not readable, so it is unclear if this statement is supported by the data.

80. Line 293: How large is the variability between subjects? Did these patterns look the same for all subjects? Again, seeing Fig. 3 would be helpful.

81. Line 295: preceding -> precede

82. Line 296: post-hoc regression analysis -> correlation analysis (should also be explained in the statistics section)

83. Line 297: Please include scatter plots showing data point for both sides and all subjects.

84. Line 301: “…this observation is not an indication that the seat motion is controlled…” – Why not?

85. Line 304: What is spinal ROM training?

86. Line 307: By… -. When…

87. Line 307: we notice -> we noticed

88. Line 311: Again, here one would need to see the activation patterns to get an impression of the timing of greater and smaller activity.

89. Line 328: Was this the case for all subjects?

90. Line 330: Why now stair walking when you compare your values to treadmill walking?

91. Line 339: This sentence is incomplete.

92. Lines 340ff: This section does not relate to the study goals. The authors should focus on discussing the results of their study.

93. Line 342: The authors did not report centre of mass speed in the results section. Please do not report additional results in the discussion section.

94. Line 345: does -> do

95. Line 355: Please define what spinal range of motion training is. Here, the authors report acute differences in the outcome parameters yet lack to evaluate the efficacy of such training. Maybe patients with LBP would not even be able to complete these tasks?

96. Line 361: This speculation goes far beyond the objective of this study. Patients with LBP often have fatty atrophy that may be caused by neurological changes rather than by changes in physical activity.

97. Line 363: Again, one would need to consider the timing of these periods of increased and reduced muscle activity.

98. Line 374: Please rephrase this sentence.

99. Line 381: What would such training look like?

100. Line 420: One limitation is that only ten subjects were enrolled in this study. Large variability in some results suggest limited generalisability of the results.

101. Lines 422ff: Again, it is likely a combination of these. To conclusively answer this question, one would need to implement a musculoskeletal model where the different parameters are altered systematically.

102. Line 427: This sentence does not relate to the results of this study.

Reviewer #2: The presented study investigated active muscle contributions during sitting on a newly designed dynamic office chair. The authors highlight the importance of new approaches to counteract prolonged (inactive) seating which is known to be associated with the occurrence of low back pain. They also acknowledge that existing “active Chair” solutions often do not lead to the intended effects, such as ease of pain occurring during prolonged sitting. Therefore the authors comprehensively investigated a newly developed “unstable” office chair, intending to promote active muscle contributions including the lower back muscles, by deploying novel movement characteristics in lateral movement directions.

The manuscript is very well written and clearly structured. The experimental design is thoroughly described, allowing to potentially replicate the performed procedures. Following a detailed presentation of the acquired results, the authors critically discuss their results and also highlight the involved limitations of this investigation.

Major issues:

- No major issues are raised for publication of the manuscript.

Minor issues/comments:

- How long did the participants on average sit on the “active chair” during testing?

- Have subjective responses/comments been assessed, on how sitting on this chair feels? Or whether participants would assume that sitting in this instable situation might be suitable for longer office working tasks?

- While the authors thoroughly discuss their results and carefully state the implications of their findings, it might be recommended to highlight that the actual benefits of this newly developed chair will need to be further tested under situations resembling more adequately real life circumstances (e.g. prolonged sitting for several hours, arm movements, changes from sitting to standing, turning while sitting,...). Furthermore its value will need to be assessed within the target population of people with low back pain.

Miscellaneous remarks:

- Line 185: a typo?: …express the RoM of an unidirectional seat motion...

- Figure caption 3: it is recommended to indicate that the data shown is “exemplary/typical data of one of the participants”

6. PLOS authors have the option to publish the peer review history of their article (what does this mean?). If published, this will include your full peer review and any attached files.

Reviewer #1: No

Reviewer #2: No

---

## [Decision Letter · Decision Letter 1]

15 Apr 2020

PONE-D-19-28644R1

Is active sitting on a dynamic office chair controlled by the trunk muscles?

PLOS ONE

Dear Mr Kuster,

Thank you for submitting your manuscript to PLOS ONE. After careful consideration, we feel that it has merit but does not fully meet PLOS ONE’s publication criteria as it currently stands. Therefore, we invite you to submit a revised version of the manuscript that addresses the points raised during the review process.

We would appreciate receiving your revised manuscript by May 30 2020 11:59PM. To enhance the reproducibility of your results, we recommend that if applicable you deposit your laboratory protocols in protocols.io, where a protocol can be assigned its own identifier (DOI) such that it can be cited independently in the future. For instructions see: http://journals.plos.org/plosone/s/submission-guidelines#loc-laboratory-protocols

We look forward to receiving your revised manuscript.

Kind regards,

Juliane Müller, PhD

Academic Editor

PLOS ONE

Additional Editor Comments (if provided):

After next round of evaluation the reviewer and I feel that the quality of your manuscript hat improved.nevertheless, there are still some language-based issues, that are recommended by one reviewer, to be revised. Please do so.

Reviewers' comments:

Reviewer's Responses to Questions

**Comments to the Author**

1. If the authors have adequately addressed your comments raised in a previous round of review and you feel that this manuscript is now acceptable for publication, you may indicate that here to bypass the “Comments to the Author” section, enter your conflict of interest statement in the “Confidential to Editor” section, and submit your "Accept" recommendation.

Reviewer #1: (No Response)

Reviewer #2: All comments have been addressed

2. Is the manuscript technically sound, and do the data support the conclusions?

Reviewer #1: Yes

Reviewer #2: Yes

3. Has the statistical analysis been performed appropriately and rigorously? 

Reviewer #1: Yes

Reviewer #2: Yes

4. Have the authors made all data underlying the findings in their manuscript fully available?

Reviewer #1: Yes

Reviewer #2: Yes

5. Is the manuscript presented in an intelligible fashion and written in standard English?

Reviewer #1: Yes

Reviewer #2: Yes

6. Review Comments to the Author

Reviewer #1: Manuscript PONE-D-19-28644

Title: Active sitting on a dynamic office chair - Is it trunk muscle controlled?

General Comments

I would like to congratulate the authors for addressing all points adequately resulting in a very clear revised manuscript. The language has much improved, however there are still some awkward sentences especially in the introduction section that require attention.

Minor Comments

Line 18: delete “, which was”

Line 19: muscle-controlled -> muscle controlled

Line 21: This was accomplished by quantifying the muscular trunk activity, thigh activity and body kinematics. -> This was accomplished by quantifying trunk and thigh muscular activity and body kinematics.

Line 38: could -> may

Line 48: It has been proposed that sitting-related LBP might be… -> Sitting-related LBP have been proposed to be…

Line 55: for the health of the spine – Please be more specific

Line 56: variations in -> altered

Line 59: some 30% to 85% -> 30 to 85%

Line 60: 34-90% -> 34 to 90%

Line 62: made great efforts to promote -> promoted

Line 68: (Fig 1), with only a small support base, to… – delete commata

Lines 102- 105: Please use lower case for all points in the list.

Lines 106-109: Delete “average” because mean is specified in brackets.

Line 109: of the -> with the

Line 100: delete “prior to the study”

Line 129: sEMG -> (sEMG) – then use only abbreviation throughout the remainder of the manuscript

Line 268ff: multifidus and iliocostalis between -> multifidus and iliocostalis muscular activity between – please also add muscular activity to all subsequent mentions of individual muscles.

Line 310: If the shift in lateral body weight controls… -> If the lateral body shift controls…

Line 339: How does active sitting affect the trunk kinematics? – Is this a subheading? If so, please correct the formatting.

Reviewer #2: The aim of this study was to investigate a newly developed dynamic office chair with a moveable seat, specifically designed to promote trunk muscle-controlled active sitting. The authors emphasize the importance of new approaches to counteract prolonged (inactive) seating. The investigations are based on the quantification of muscular trunk/thigh activity and related body kinematics. The results are clearly presented and critically discussed afterwards.

The manuscript was given a comprehensive revision. It is well written and still very clearly structured. All main comments/questions of the previous submission have been addressed.

One minor comment

Line 102: inclusion criteria for height between 1.53 and 1.92 cm. Was this an inclusion criterion or it just happened to be the range of the respective participants?

7. PLOS authors have the option to publish the peer review history of their article (what does this mean?). If published, this will include your full peer review and any attached files.

Reviewer #1: No

Reviewer #2: No

---

## [Author Response · Author response to Decision Letter 1]

29 Apr 2020

See attached point-by-point response file (Response to Reviewers.docx)

---

## [Decision Letter · Decision Letter 2]

11 Nov 2020

Is active sitting on a dynamic office chair controlled by the trunk muscles?

PONE-D-19-28644R2

Dear Dr. Kuster,

We’re pleased to inform you that your manuscript has been judged scientifically suitable for publication and will be formally accepted for publication once it meets all outstanding technical requirements.

Kind regards,

Juliane Müller, PhD

Academic Editor

PLOS ONE

Additional Editor Comments (optional):

no further

Reviewers' comments:

Reviewer's Responses to Questions

**Comments to the Author**

1. If the authors have adequately addressed your comments raised in a previous round of review and you feel that this manuscript is now acceptable for publication, you may indicate that here to bypass the “Comments to the Author” section, enter your conflict of interest statement in the “Confidential to Editor” section, and submit your "Accept" recommendation.

Reviewer #1: All comments have been addressed

Reviewer #2: All comments have been addressed

2. Is the manuscript technically sound, and do the data support the conclusions?

Reviewer #1: Yes

Reviewer #2: Yes

3. Has the statistical analysis been performed appropriately and rigorously? 

Reviewer #1: Yes

Reviewer #2: Yes

4. Have the authors made all data underlying the findings in their manuscript fully available?

Reviewer #1: Yes

Reviewer #2: Yes

5. Is the manuscript presented in an intelligible fashion and written in standard English?

Reviewer #1: Yes

Reviewer #2: Yes

6. Review Comments to the Author

Reviewer #1: I congratulate you on this very nice manuscript on a very interesting study and novel results with many implications.

Reviewer #2: All comments/questions of the previous submission have been addressed.

The work has further been tweaked by minor changes all over the manuscript.

7. PLOS authors have the option to publish the peer review history of their article (what does this mean?). If published, this will include your full peer review and any attached files.

Reviewer #1: No

Reviewer #2: No

---

## [Editor Report · Acceptance letter]

17 Nov 2020

PONE-D-19-28644R2 

Is active sitting on a dynamic office chair controlled by the trunk muscles?

Dear Dr. Kuster:

I'm pleased to inform you that your manuscript has been deemed suitable for publication in PLOS ONE. Congratulations! Your manuscript is now with our production department. 

Kind regards, 

on behalf of

Dr. Juliane Müller 

Academic Editor

PLOS ONE